# Indirect Treatment Comparison of Larotrectinib versus Entrectinib in Treating Patients with TRK Gene Fusion Cancers

**DOI:** 10.3390/cancers14071793

**Published:** 2022-03-31

**Authors:** Jesus Garcia-Foncillas, Carsten Bokemeyer, Antoine Italiano, Karen Keating, Noman Paracha, Marc Fellous, Marisca Marian, Mirko Fillbrunn, Wei Gao, Rajeev Ayyagari, Ulrik Lassen

**Affiliations:** 1Department of Oncology, Cancer Institute, Fundacion Jimenez Diaz University Hospital, Autonomous University, 28040 Madrid, Spain; 2Department of Oncology, Hematology and BMT with Section Penumology, University Medical Center Hamburg—Eppendorf, 20246 Hamburg, Germany; cbokemeyer@uke.de; 3Early Phase Trials Unit, Institut Bergonie, 33000 Bordeaux, France; a.italiano@bordeaux.unicancer.fr; 4Faculty of Medicine, University of Bordeaux, 33000 Bordeaux, France; 5Bayer HealthCare Pharmaceuticals, Inc., Whippany, Hanover Township, NJ 07981, USA; karen.keating@bayer.com; 6Market Access Oncology, Bayer AG, 4052 Basel, Switzerland; noman.paracha@bayer.com (N.P.); marc.fellous@bayer.com (M.F.); marisca.marian@bayer.com (M.M.); 7Analysis Group, Inc., Boston, MA 02199, USA; mirko.fillbrunn@analysisgroup.com (M.F.); wei.gao@analysisgroup.com (W.G.); rajeev.ayyagari@analysisgroup.com (R.A.); 8Department of Oncology, Rigshospitalet, 2100 Copenhagen, Denmark; ulrik.lassen@regionh.dk

**Keywords:** larotrectinib, entrectinib, NTRK gene fusion, safety, clinical efficacy

## Abstract

**Simple Summary:**

Larotrectinib and entrectinib have never been directly compared in a clinical trial for the treatment of TRK fusion-positive cancer, so a comparison must use separate data from each drug’s trials. This study used established statistical methods to balance the patient populations across trials and found that, compared to entrectinib, larotrectinib was associated with a higher overall survival, longer duration of response, and higher complete response rates, and numerically better progression-free survival and similar overall response and safety rates. Based on treatment guidelines, healthcare stakeholders have only one opportunity to decide which TRK inhibitor to select for patients. The results of this analysis can help physicians decide between available treatment options for TRK fusion-positive solid cancer.

**Abstract:**

Information regarding the comparative efficacy of first-generation receptor tyrosine kinase inhibitors is limited. This matching-adjusted indirect comparison (MAIC) evaluated differences in efficacy and safety across larotrectinib and entrectinib trials. Data from clinical trials for larotrectinib (LOXO-TRK-14001 (NCT02122913), SCOUT (NCT02637687), and NAVIGATE (NCT02576431)) and entrectinib (ALKA-372-001 (EudraCT 2012-000148-88), STARTRK-1 (NCT02097810), and STARTRK-2 (NCT02568267)) were used. Adults (≥18 years) across trials were matched on available baseline characteristics. Outcomes evaluated included overall response rate (ORR), complete response (CR) rate, duration of response (DoR), overall survival (OS), progression-free survival (PFS), any serious treatment-related adverse events of grade ≥ 3 (TRAEs), and TRAEs leading to treatment discontinuation. The MAIC included 74 patients from entrectinib trials and 117 and 147 patients for the larotrectinib efficacy and safety populations, respectively. Post-matching, larotrectinib was associated with a significantly longer median duration of OS than entrectinib (*p* < 0.05) and a numerically longer median PFS (*p* = 0.07). ORR was similar for both agents (*p* = 0.63). The CR rate was higher (*p* < 0.05) and the DoR was longer for larotrectinib (*p* < 0.05). Safety outcomes were comparable and low for both treatments. Results were consistent in sensitivity analyses. These findings suggest favorable efficacy for larotrectinib and comparable safety profiles versus entrectinib in treating tropomyosin receptor kinase fusion cancer.

## 1. Introduction

Neurotrophic tyrosine receptor kinases (NTRKs) comprise a family of genes that encode tropomyosin receptor kinases (TRKs), which are involved in the development and maintenance of the nervous system [1,2]. Gene fusion events that involve *NTRK* genes (*NTRK1*, *NTRK2*, and *NTRK3*) can promote overexpression of TRK receptors leading to tumorigenesis [3]. Among adult and pediatric patients, the incidence of *NTRK* gene fusions varies between <5% in solid cancers (e.g., lung cancer, colorectal cancer, glioma) and >75% in rare cancers (e.g., infantile fibrosarcoma, secretory breast, secretory carcinoma of the salivary gland) [4]. The presence of *NTRK* gene fusions in both adult and pediatric populations suggests it may be one of the first oncogenic drivers that are both tissue- and age-agnostic [5]. Despite the rarity of *NTRK* gene fusions among certain tumor types, the development of TRK inhibitors has been an active area of research [6].

Previously, treatment options for TRK cancers were limited to chemotherapy, biologic therapy, or immunotherapy [7]. The development of targeted agents has dramatically improved the treatment landscape for patients harboring *NTRK* gene fusions [3,8,9,10,11]. In 2018, the United States Food and Drug Administration (FDA) approved larotrectinib, a highly specific inhibitor of all three TRK proteins, for adult and pediatric patients with solid tumors [12]. Larotrectinib is indicated for patients with solid tumors with confirmed *NTRK* gene fusion without a known acquired resistance mutation, who have metastatic cancer or would experience severe morbidity following surgical resection, and who show evidence of progression following treatment or have no adequate treatment alternative [12]. Larotrectinib’s approval was based on results from three multi-center clinical studies (a phase 1 trial (NCT02122913), SCOUT (NCT02637687), and NAVIGATE (NCT02576431)) [5,13]. In these studies, larotrectinib exhibited a response rate of 75% (according to independent review) in patients with TRK fusion-positive tumors across 17 different cancer types, with 71% of responses maintained one year following treatment initiation [5]. In 2019, entrectinib, a multi-kinase inhibitor (targets include TRK proteins, c-ros oncogene 1 (ROS1), and anaplastic lymphoma kinase (ALK) [14]), was approved by the FDA for adult and pediatric patients (≥12 years) with the same indication as larotrectinib [15]. Entrectinib’s approval was based on results from the following multi-center, single-arm trials: ALKA-372-001 (EudraCT 2012-000148-88), STARTRK-1 (NCT02097810), and STARTRK-2 (NCT02568267) [9,10,11]. Results from these trials revealed a response rate of 57% in patients with TRK fusion-positive solid tumors across 10 different tumor types [9,10,11].

Despite the demonstrated efficacy of larotrectinib and entrectinib in their respective trials [5,9,10,11,13], insight into their comparative efficacy and safety is lacking without a direct head-to-head comparison [16]. Simply comparing outcomes between larotrectinib and entrectinib may be subject to significant biases confounded by differences in study designs and trial populations. To overcome these challenges, an indirect assessment of the comparative efficacy and safety between pivotal trials of larotrectinib and entrectinib while accounting for cross-trial differences is needed. The present study utilized a widely used indirect treatment comparison approach, matching-adjusted indirect comparison (MAIC) [17,18], to fill the knowledge gap. MAIC has been applied in multiple disease areas to support reimbursement submissions and publications and has been recognized in guidance from the National Institute for Health and Care Excellence (NICE) [19].

## 2. Materials and Methods

### 2.1. Data Sources

Patient-level data for larotrectinib was based on the integrated patient population harboring TRK fusion-positive tumors across three clinical trials: larotrectinib’s phase 1 trial, SCOUT, and NAVIGATE. The phase 1 trial, a multi-center phase 1 study, assessed dose escalation of larotrectinib (50–200 mg) with advanced solid tumors (including both TRK-positive and -negative patients) to evaluate its safety and pharmacokinetics [13]. SCOUT, a phase 1/2 multi-center study, assessed the safety of larotrectinib (50–200 mg) with advanced solid or primary central nervous system (CNS) tumors [5]. NAVIGATE, a phase 2 multi-center basket study of larotrectinib (100 mg), investigated its efficacy in the treatment of advanced solid tumors harboring a fusion of *NTRK* types 1–3 in children and adults [5]. Specific eligibility criteria varied across the three trials as reported previously [5,13]. The combined larotrectinib trial population included patients recruited worldwide. Larotrectinib was administered as monotherapy once or twice daily until either disease progression, withdrawal of the patient from the study, or the occurrence of an unacceptable level of adverse events. The July 2020 data cut for the integrated patient population were used in the current analysis.

For entrectinib, aggregate results for the integrated patient population were extracted from three phase 1/2 clinical trials: ALKA-372-001, STARTRK-1, and STARTRK-2 [9,10,11]. ALKA-372-001 was a phase 1, multi-center study assessing entrectinib’s dosing schedule in patients with advanced or metastatic solid tumors with TRKA/B/C, ROS1, or ALK gene fusions in Italy among patients treated with entrectinib. STARTRK-1 was a phase 1, multi-center study of oral entrectinib in adult patients with locally advanced or metastatic cancer confirmed to be positive for *NTRK1*, *NTRK2*, *NTRK3*, ROS1, or ALK gene fusions among patients treated with entrectinib. STARTRK-2 was a phase 2, global, multi-center basket study in patients with solid tumors that harbor an *NTRK1/2/3*, ROS1, or ALK-positive gene fusion. Summary statistics were obtained from the literature (data cutoff: 31 October 2018) [9] with supplementary information from an earlier publication (data cutoff: 31 May 2018) [10] and FDA prescribing information [15]. The combined entrectinib trial population included patients recruited worldwide. Entrectinib was administered as monotherapy intermittently or once daily until either documented radiographic progression, unacceptable toxicity, or withdrawal of consent.

### 2.2. Sample Selection

Key inclusion/exclusion criteria from entrectinib trials were applied to larotrectinib’s individual patient data to increase the comparability across trials before matching on the selected baseline characteristics. This analysis did not include patients with primary central nervous system tumors as the efficacy population of the entrectinib trials did not include patients with primary central nervous system tumors. After selection of patients in the efficacy or safety population, the following criteria were applied:Documented *NTRK* fusion as determined by an independent radiology committee;Patients were required to be 18 years or older;Patients were required to have Eastern Cooperative Oncology Group (ECOG) scores of 2 or less;Patients were required to be TRK inhibitor naïve.

### 2.3. Outcome Measures

#### 2.3.1. Efficacy Outcomes

Efficacy outcomes examined in the current analysis included overall survival (OS), progression-free survival (PFS), duration of response (DoR), overall response rate (ORR), and complete response (CR) rate. OS was defined as the time from the first dose to death due to any cause. PFS was defined as the time from the first dose to the first documented radiographic disease progression or death due to any cause, whichever occurred first (according to an independent central review). DoR was defined as the time from the date of first CR/partial response (PR) to the first documented radiographic disease progression or death due to any cause, whichever occurred first. ORR was defined as the proportion of patients with a CR/PR as assessed by Response Evaluation Criteria in Solid Tumors (RECIST) version 1.1 [20] or Response Assessment in Neuro-Oncology Brain Metastases (RANO) criteria [21]. CR rate was defined as the proportion of patients with a CR as assessed by criteria in RECIST version 1.1 or RANO.

In the entrectinib trials, tumor assessments to be used for PFS, DoR, CR, and ORR were performed when clinical deterioration was suspected or at scheduled assessments (at the end of cycle 1 (4 weeks) and at the end of alternate cycles thereafter (i.e., every 8 weeks), when a clinical deterioration occurred, and at the end of treatment if not performed in the previous 4 weeks). In the larotrectinib trials, tumor assessments were taken at baseline and every 2 cycles (8 weeks) up to 1 year, and every 3 cycles thereafter until disease progression.

Time-to-event outcomes in the entrectinib trials were extracted from the published Kaplan–Meier curves using digitization software (WebPlotDigitizer, https://automeris.io/WebPlotDigitizer, accessed on 20 June 2020). Based on the extracted curves and reported numbers of events and patients at risk at various time points, pseudo-patient-level data of these time-to-event outcomes were generated using the Guyot method [22]. ORR and CR were extracted as reported in the published literature for entrectinib.

#### 2.3.2. Safety Outcomes

Safety outcomes examined in the analysis included any serious treatment-related adverse event (TRAE) and TRAE that led to discontinuation. Any serious treatment-related adverse event (TRAE) was defined as the proportion of patients with TRAEs of grade 3 or higher. TRAE that led to discontinuation was defined as the proportion of patients with a TRAE leading to treatment discontinuation.

For larotrectinib, adverse events were assessed from the date that informed consent was obtained until at least 28 days following administration of the last dose. For both treatments, adverse events were coded using the Medical Dictionary for Regulatory Activities, and classified and graded using the Common Terminology Criteria for Adverse Events version 4.03 [23].

### 2.4. Statistical Analysis

Single-arm MAICs were conducted to compare feasible outcomes between larotrectinib and entrectinib in treating TRK fusion-positive solid cancers because of the single-arm trial design for larotrectinib and entrectinib. Patient-level data from larotrectinib and aggregate results from entrectinib (see Section 2.1) were used in the analysis. Patients treated with larotrectinib were assigned weights so that the weighted average of selected baseline characteristics matched those of the entrectinib patient population. The weights were obtained based on a logistic regression model for the propensity of enrollment in the larotrectinib trials vs. the entrectinib trials. Because only summary data were available for the entrectinib trials, the logistic regression model was estimated using the method of moments [24].

After matching, baseline characteristics and outcomes of interest were compared between the weighted larotrectinib trial population and the entrectinib trial population using weighted *t*-tests for continuous variables and weighted chi-squared tests for categorical variables. Differences among continuous outcomes and risk differences (RDs) for categorical outcomes were calculated. Robust sandwich estimators were used to estimate standard errors, which provide valid estimates for 95% confidence intervals (CIs; i.e., achieved their nominal coverage probability). For time-to-event outcomes, survival curves were compared using weighted log-rank tests and hazard ratios (HRs) were estimated from weighted, otherwise unadjusted Cox proportional hazards models.

Separate weights were estimated for the efficacy and safety populations. As no separate baseline characteristics were reported for entrectinib’s safety population, the analysis assumed that the average baseline characteristics of the safety population were the same as that of the efficacy population.

In the primary analysis, the matching baseline characteristics included sex, age (>57 years; to match the median age range in the entrectinib population), race (White, Black, Asian, Other), ECOG score (0, 1, 2), tumor type (thyroid, sarcoma, salivary, lung), metastatic disease (vs. locally advanced, unresectable disease), *NTRK* fusion type (*NTRK1*, *NTRK2*, *NTRK3*), prior lines of systemic therapy for metastatic disease (0, 1, 2, 3+), and previous central nervous system metastases. These baseline clinical and demographic characteristics were selected based on their availability in both trial populations and clinical input on their potential to be treatment effect modifiers.

Three sensitivity analyses were conducted, matching on the same variables as the primary analysis except (1) adding gastrointestinal tumors and (2) replacing the number of prior lines of therapy by the type of prior treatments (hormonal therapy, chemotherapy, immunotherapy, and targeted therapy).

A third sensitivity analysis was conducted using the simulated treatment comparison (STC) method to assess the robustness of the results to a different indirect treatment comparison method [25]. The STC first included regression models of the outcomes on the same set of the matching baseline characteristics in the primary analysis using the larotrectinib patient-level data. Cox regression models were used for OS, PFS, and DoR; logistic regression models were used for ORR, CR, and the safety outcomes. The STC then combined the regression model estimates, reported baseline characteristics for the entrectinib trial population, and correlations between covariates estimated from larotrectinib’s patient-level data to simulate pseudo-patient-level outcome data for a hypothetical larotrectinib-treated population in the entrectinib trial. The simulated outcome data were then compared against the reported outcomes for entrectinib. An STC relies on different assumptions compared to the MAIC; in comparison, it is less sensitive to missing overlap between studies, but requires correct specification of the underlying model. These differences in assumptions therefore provide an additional test of the robustness of the analyses conducted here.

Statistical significance was considered at the 5% level. All analyses were conducted using SAS Enterprise Version 7.15 (SAS Institute, Cary, NC, USA) and R Version 3.6.3.

## 3. Results

### 3.1. Baseline Characteristics

A total of 331 patients from the three larotrectinib trials were assessed for inclusion in the MAIC. Of these patients, TRK fusion-positive tumors were present among 192 patients in the independent review committee-assessed efficacy population. After applying the sample selection criteria, 117 patients were included in the larotrectinib efficacy population used in the analyses. The respective numbers for the TRK fusion-positive safety population, defined as the patients who received at least one dose of treatment, were 260 and 147 patients.

The baseline characteristics before and after matching in the primary analysis are summarized in Table 1 (efficacy population) and Table 2 (safety population). Baseline characteristics were reported among 74 patients with TRK fusion-positive tumors in the entrectinib trials. One baseline characteristic, metastatic disease, was only available from an earlier data cut of entrectinib (31 May 2018) comprising 54 patients.

Baseline characteristics in the efficacy populations were similar across trials before and after matching. Before matching, slightly less than half of included patients were male (46.2% for larotrectinib and 47.3% for entrectinib, *p* > 0.99), 44.4% of larotrectinib patients were younger than 57 years (*p* = 0.55), the median of age for entrectinib, and most patients were White (73.5% for larotectinib, 70.3% for entrectinib, respectively, *p* = 0.75).

The majority of patients had an ECOG score of 0 (35.0%, 40.5%, resp., *p* = 0.54) or 1 (52.1%, 45.9%, resp., *p* = 0.49). Thyroid tumor, sarcoma, salivary cancer, lung cancer, and gastrointestinal cancer were common primary cancer types across the trials and had similar frequency between the trials. Most patients had metastatic disease (90.6%, 96.3%, resp., *p* = 0.23) at baseline, and *NTRK1* (44.4%, 40.5%, resp., *p* = 0.70) and *NTRK3* (53.0%, 56.8%, resp., *p* = 0.72) were the most common *NTRK* gene fusions. While no patients with primary central nervous system tumors were included in the analysis, a total of 12.0% and 21.6% of patients in the respective trials had central nervous metastases at baseline (*p* = 0.11). The proportion of patients without any prior lines of systemic therapy for metastatic disease was similar between the trials (26.5%, 27.0%, resp., *p* > 0.99); fewer patients had 1 or 2 prior lines but more patients had 3+ lines in the larotrectinib trials compared to patients in the entrectinib trials (25.2% (larotrectinib) vs. 28.4% (entrectinib), 20.4% vs. 27.0%, and 27.9% vs. 17.6%, resp.; all comparisons were non-significant). The proportion of patients with prior chemotherapy was statistically different before matching and remained so afterwards (58.1% before matching, 81.1% after matching, *p* < 0.01). The proportion of patients with prior hormonal therapy in the larotrectinib trial population was numerically lower compared to entrectinib (5.1%, 12.2%, resp., *p* = 0.14), while type of prior therapy for immunotherapy, and targeted therapy were similar between the two treatments.

After matching, the summary statistics for the baseline characteristics that were included in the matching were the same between the trial populations due to the nature of the matching process; results of the analyses are reported in Table 3 and Table 4. For type of prior therapy, which was not included in the matching process, differences remained after matching. The proportion of patients with prior chemotherapy remained statistically different after matching (58.1%, 81.1%, resp., *p* < 0.01) and the proportion of patients with prior hormonal therapy became statistically different after matching (2.0%, 12.2%, resp., *p* < 0.01). While the available sample size prevented the use of both number of prior lines of therapy and type of prior therapy, a sensitivity analysis replaced number of prior lines of therapy with prior type of therapy; results are reported in Table 5. Baseline characteristics in the safety population were generally similar to the efficacy population.

### 3.2. Efficacy Outcomes

RDs and HRs with corresponding 95% CIs used to assess efficacy outcomes between larotrectinib and entrectinib before and after matching are summarized in Table 3.

Prior to matching, the median OS for larotrectinib was not reached ((95% CI): (40.7, not evaluable (NE))) and the median OS for entrectinib was 23.9 months (16.0, NE). Larotrectinib had longer OS compared to entrectinib, with an HR comparing the two treatments of 0.43 (0.24, 0.76) (*p* < 0.01) in favor of larotrectinib. After matching, the median OS for larotrectinib was not reached (38.7, NE) and larotrectinib had longer OS compared to entrectinib, with an HR of 0.43 (0.23, 0.83) (*p* < 0.05) in favor of larotrectinib (Figure 1).

Larotrectinib had a median PFS of 33.0 months (16.6, NE) before matching and entrectinib had a median PFS of 11.2 (8.0, 15.7). Larotrectinib patients had significantly longer PFS compared to entrectinib patients with an HR of 0.56 (0.37, 0.86) (*p* < 0.01). After matching, larotrectinib had a median PFS of 19.3 (11.1, 55.7) and demonstrated a numerically longer PFS compared to entrectinib with an HR of 0.66 (0.42, 1.03) (*p* = 0.07) (Figure 2).

Before matching, the ORR was 65.0% for larotrectinib and 63.5% for entrectinib, resulting in an RD of 1.5% (−12.5%, 15.4%) (*p* = 0.84). After matching, the ORR was 67.3% for larotrectinib, resulting in an RD of 3.8% (−11.7%, 19.3%) (*p* = 0.63). The CR rate was significantly higher for larotrectinib with 19.7% versus 6.8% for entrectinib before matching, resulting in an RD of 12.9% (3.7%, 22.1%) (*p* < 0.01). After matching, larotrectinib persisted to have a higher CR with 20.3%, resulting in an RD of 13.5% (2.9%, 24.1%) (*p* < 0.05).

Median DoR for larotrectinib was 41.5 months (32.5, NE) versus 12.9 (9.3, NE) for entrectinib before matching. Comparing the two treatments, DoR was significantly longer for larotrectinib with an HR of 0.33 (0.17, 0.63) (*p* < 0.001) compared to entrectinib. After matching, DoR for larotrectinib was 32.5 (17.4, NE) and DoR remained significantly longer for larotrectinib with an HR of 0.49 (0.25, 0.98) (*p* < 0.05) (Appendix A).

The results of the sensitivity analyses for the assessment of the efficacy outcomes between larotrectinib and entrectinib before and after matching were similar to the results of the primary analysis, both for the different specifications of the MAIC as well as the STC results (Table 5).

### 3.3. Safety Outcomes

The RDs between larotrectinib and entrectinib before and after matching in the assessment of safety outcomes are summarized in Table 4. Before matching, larotrectinib was associated with numerically lower rates of serious TRAEs (RD: −4.6%; *p* = 0.27) and TRAEs leading to discontinuation (RD: −3.3%; *p* = 0.18) compared to entrectinib. Results were similar between treatments after matching (serious TRAEs RD: −3.7%, *p* = 0.40; TRAEs leading to discontinuation RD: −3.3%, *p* = 0.18). Safety rates remained similar between larotrectinib and entrectinib in the sensitivity analyses (Table 5).

## 4. Discussion

This indirect treatment comparison provides clinicians and healthcare stakeholders with key insights when considering larotrectinib and entrectinib in the absence of a head-to-head randomized clinical trial. Before and after adjusting for baseline differences, larotrectinib demonstrated a favorable efficacy profile and comparable safety profile compared to entrectinib. After adjusting for baseline differences between patient populations, larotrectinib was associated with significantly better OS, CR, and DoR compared to entrectinib. In addition, larotrectinib was associated with numerically better ORR and PFS. Sensitivity analyses showed similar results as the primary analysis, which supports the robustness of the results. Particularly, in the sensitivity analysis accounting for differences in prior treatment categories between the larotrectinib and entrectinib trial populations, larotrectinib also demonstrated significantly longer PFS compared to entrectinib.

Although both entrectinib and larotrectinib are reported to be potent TRK inhibitors, differences in their respective mechanisms of action may help to explain the differences in their comparative efficacy. Entrectinib is a multi-kinase inhibitor that targets the three TRK proteins in addition to *ROS1* and *ALK* [14]. In contrast, larotrectinib, which is currently the most specific TRK inhibitor, is a selective inhibitor of the three TRK proteins [5,26]. Despite earlier discussion that multi-targeted agents may confer greater clinical benefits compared to single targeted agents [27], this difference in their mechanism of action could explain the observed differences in efficacy between the two treatments.

After adjusting for baseline differences, larotrectinib demonstrated similar safety outcomes when compared to entrectinib, both of which had low rates of TRAEs. As the clinical management of cancer improves, the impact of TRAEs is of increasing importance to healthcare stakeholders. Much of this is due to the impact of TRAEs on patients’ quality of life, compliance with clinical regimens, and the clinical and economic burden of illness [28,29]. For example, a recent retrospective analysis found that roughly 20% of unscheduled hospitalizations among patients with solid tumors were due to TRAEs [29]. Although interventions to identify and mitigate the impact of TRAEs can help address this challenge, these efforts may result in an increase in resource utilization that can increase the economic burden associated with the disease. As a result, the use of targeted agents with tolerable safety profiles represents a key strategy to mitigate negative outcomes from TRAEs.

Overall, the findings from the present indirect treatment comparison fill an important knowledge gap regarding the impact of treatment with larotrectinib and entrectinib. To the best of our knowledge, only one study to date has sought to provide a comparative analysis of larotrectinib and entrectinib [16]. In that study, a partitioned survival model was developed to compare differences in expected life-years and quality-adjusted life-years for larotrectinib versus entrectinib in the second line of therapy for TRK fusion-positive metastatic non-small cell lung cancer [16]. Results showed that in the base case, treatment with larotrectinib resulted in 5.4 median preprogression life-years (7.0 median total life-years) versus 1.2 median preprogression life-years (1.8 median total life-years) with entrectinib [16]. Although the survival gains resulting from treatment with larotrectinib were not confirmed in their analysis, scenario analyses suggested that gains may persist. The benefits associated with larotrectinib compared to entrectinib in that study align with findings from the present study’s demonstration that larotrectinib is associated with superior efficacy and comparable safety profiles compared to entrectinib.

Often times, clinicians and other healthcare stakeholders have only one opportunity to decide which TRK inhibitor to select for patients due to the risk of acquired resistances [6]. This study has the potential to aid healthcare professionals and clinical decision makers when considering which treatment would be more appropriate for their patients. With the current study offering healthcare stakeholders the opportunity to make timely comparisons in the absence of head-to-head randomized trials, patients may receive essential treatment sooner, which can result in clear clinical gains.

## 5. Limitations

The findings from this study should be considered within the context of certain limitations. First, due to the rarity of *NTRK* fusions, the available sample size from the relevant clinical trials was limited. As a result, the ability to adjust for prior lines of therapy and prior types of treatment concurrently within the same analysis was limited. However, the results separately adjusting for prior lines of therapy and prior types of treatment were comparable, which validates the robustness of the current study. Second, the necessary information for an adjusted comparison (baseline characteristics and analysis results) were not reported by tumor type or other subgroups of interest for entrectinib, which prevented adjusted subgroup analyses. While the current study used the pooled population and matched on the tumor types, the adjustment could only be made for the most common tumor types due to the limited sample size. Third, follow-up time differed between the two treatments, with longer follow-up observed in larotrectinib. While the analyses of time-to-event efficacy outcomes (OS and PFS) explicitly account for different follow-up times and are thus robust to differences in follow-up, the analysis of other outcomes may be affected by different follow-up times. However, the difference in median follow-up between treatments was less than 3 months compared to overall follow up of 14 to 16 months, suggesting that sufficient follow-up was reached for both treatments. Fourth, without a common comparator arm for the outcome comparisons, it was not feasible to validate whether the adjustments fully balance the characteristics of the study populations. However, MAIC and STC have been established as valid approaches for comparing single-arm trials [19]. Finally, while MAIC and STC adjust for differences in baseline characteristics that are available and similarly measured across trial populations, the comparisons may be biased by differences in unobserved baseline characteristics that affect outcomes. Adjusting for additional baseline characteristics not yet included in the analysis could further improve the estimate of the relative treatment effects. The current analysis already includes a wide range of key baseline characteristics, and so it is unclear whether additional baseline characteristics may significantly affect the results presented here. Ultimately, only a well-conducted, head-to-head randomized trial comparison can avoid the potential bias due to unobserved baseline differences.

## 6. Conclusions

The clinical trial data used in this indirect treatment comparison suggest that the mechanism of NTRK inhibition is efficacious and safe in treating solid cancers with TRK fusion-positive cancer, and the present indirect treatment comparison showed that larotrectinib had a favorable efficacy and comparable safety profile compared to entrectinib after adjusting for the heterogeneity in patient characteristics between the trial populations. These findings can help to inform clinicians and other healthcare stakeholders when considering which treatment option would provide the greatest clinical benefit to patients. Further analyses are suggested to confirm these findings.

## Figures and Tables

**Figure 1 cancers-14-01793-f001:**
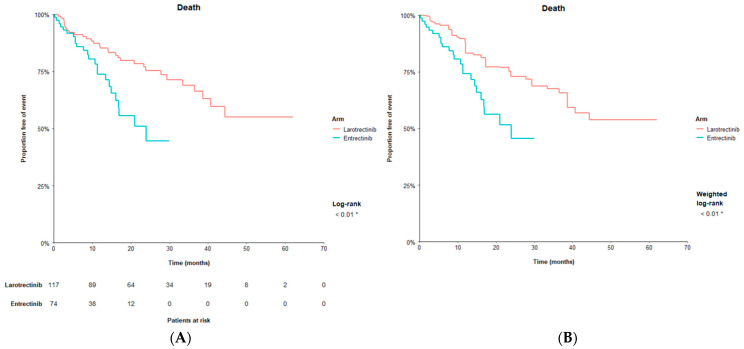
Overall survival with larotrectinib vs. entrectinib. (**A**) Kaplan–Meier curve for overall survival before matching. (**B**) Kaplan–Meier curve for overall survival after matching on primary matching factors ^a^. Notes: ^a^^.^ Primary matching variables include the following: male, age above median in entrectinib population in Rolfo, 2020 [9] (>57 years), White, Black, Asian, ECOG PS score 0, ECOG PS score 1, tumor (thyroid), tumor (sarcoma), tumor (salivary), tumor (lung), metastatic disease (vs. locally advanced, unresectable disease), central nervous system metastases (yes), *NTRK1*, *NTRK2*, prior lines of systemic therapy for metastatic disease (0), prior lines of systemic therapy for metastatic disease (1), prior lines of systemic therapy for metastatic disease (2). * denotes statistical significance (alpha < 0.05).

**Figure 2 cancers-14-01793-f002:**
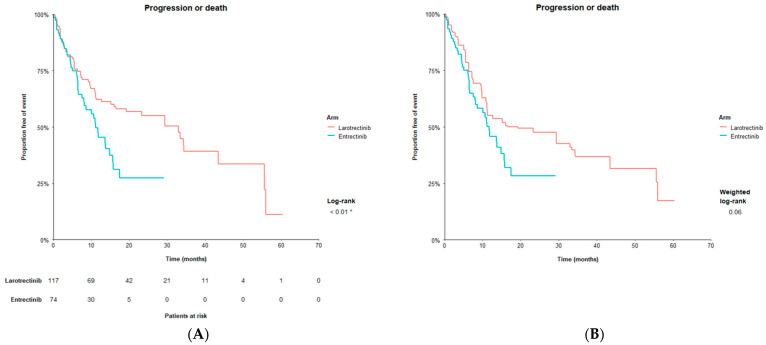
Progression-free survival with larotrectinib vs. entrectinib. (**A**) Kaplan–Meier curve for progression-free survival before matching. (**B**) Kaplan–Meier curve for progression-free survival after matching on primary matching factors ^a^. Notes: ^a.^ Primary matching variables include the following: male, age above median in entrectinib population in Rolfo [9], 2020 (>57 years), White, Black, Asian, ECOG PS score 0, ECOG PS score 1, tumor (thyroid), tumor (sarcoma), tumor (salivary), tumor (lung), metastatic disease (vs. locally advanced, unresectable disease), central nervous system metastases (yes), *NTRK1, NTRK2*, prior lines of systemic therapy for metastatic disease (0), prior lines of systemic therapy for metastatic disease (1), prior lines of systemic therapy for metastatic disease (2). * denotes statistical significance (alpha < 0.05).

**Table 1 cancers-14-01793-t001:** Baseline characteristics before and after matching of larotrectinib efficacy population and entrectinib.

Variables	Entrectinib *N* = 74	Larotrectinib *N* = 117		
		Before Matching		After Matching ^a^
	*N* (%)	*N* (%)	*p*-Value ^b^	%
Male	35 (47.3%)	54 (46.2%)	>0.99	47.3%
Age above 57 years	37 (50.0%)	52 (44.4%)	0.55	50.0%
Race				
White	52 (70.3%)	86 (73.5%)	0.75	70.3%
Black	2 (2.7%)	5 (4.3%)	0.71	2.7%
Asian	13 (17.6%)	14 (12.0%)	0.38	17.6%
Other/Not reported	7 (9.5%)	12 (10.3%)	>0.99	9.4%
ECOG PS score				
0	30 (40.5%)	41 (35.0%)	0.54	40.5%
1	34 (45.9%)	61 (52.1%)	0.49	45.9%
2	10 (13.5%)	15 (12.8%)	>0.99	13.6%
Primary tumor type				
Thyroid	7 (9.5)	25 (21.4)	0.05	9.5
Salivary	13 (17.6)	21 (17.9)	>0.99	17.6
Sarcoma	16 (21.6)	25 (21.4)	>0.99	21.6
Lung	13 (17.6)	13 (11.1)	0.29	17.6
Other	25 (33.8)	33 (28.2)	0.51	33.7
Metastatic disease (vs. locally advanced, unresectable disease)	52 (96.3)	106 (90.6)	0.23	96.3
Central nervous system metastases (Yes)	16 (21.6)	14 (12.0)	0.11	21.6
*NTRK* gene fusion				
*NTRK1*	30 (40.5)	52 (44.4)	0.70	40.5
*NTRK2*	2 (2.7)	3 (2.6)	>0.99	2.7
*NTRK3*	42 (56.8)	62 (53.0)	0.72	56.8
Prior lines of systemic therapy for metastatic disease				
0	20 (27.0)	30 (25.6)	0.97	27.0
1	21 (28.4)	29 (24.8)	0.70	28.4
2	20 (27.0)	23 (19.7)	0.31	27.0
3+	13 (17.6)	35 (29.9)	0.08	17.6
Prior therapy (chemotherapy) ^c^	60 (81.1)	68 (58.1%)	<0.01 *	58.1
				*p* < 0.01 *
Prior therapy (hormonal therapy) ^c^	9 (12.2)	6 (5.1%)	0.14	2.0
				*p* < 0.01 *
Prior therapy (immunotherapy) ^c^	9 (12.2)	14 (12.0%)	1.00	15.9
				*p* = 0.53
Prior therapy (targeted therapy) ^c^	18 (24.3)	31 (26.5%)	0.87	18.5
				*p* = 0.36

^a^ Matching variables include the following: male, age above median in entrectinib population in Rolfo, 2020 (>57 years), White, Black, Asian, ECOG PS score 0, ECOG PS score 1, tumor (thyroid), tumor (sarcoma), tumor (salivary), tumor (lung), metastatic disease (vs. locally advanced, unresectable disease), central nervous system metastases (yes), *NTRK1*, *NTRK2*, prior lines of systemic therapy for metastatic disease (0), prior lines of systemic therapy for metastatic disease (1), prior lines of systemic therapy for metastatic disease (2). ^b^
*p*-values for continuous variables were calculated using the Wald test. *p*-values for categorical variables were calculated using the Wald test. ^c^ In the primary analysis, prior therapy type was not adjusted for. * denotes statistical significance (alpha < 0.05). ECOG: Eastern Cooperative Oncology Group; *NTRK*: neurotrophic receptor tyrosine kinase; PS: performance status.

**Table 2 cancers-14-01793-t002:** Baseline characteristics before and after matching of larotrectinib safety population and entrectinib.

Variables	Entrectinib *N* = 74	Larotrectinib *N* = 147		
		Before Matching		After Matching ^a^
	*N* (%)	*N* (%)	*p*-Value ^b^	%
Male	35 (47.3)	71 (48.3)	>0.99	47.3
Age above 57 years	37 (50.0)	62 (42.2)	0.34	50.0
Race				
White	52 (70.3)	95 (64.6)	0.49	70.3
Black	2 (2.7)	5 (3.4)	>0.99	2.7
Asian	13 (17.6)	33 (22.4)	0.51	17.6
Other/Not reported	7 (9.5)	14 (9.5)	>0.99	9.4
ECOG PS score				
0	30 (40.5)	54 (36.7)	0.69	40.5
1	34 (45.9)	74 (50.3)	0.63	45.9
2	10 (13.5)	19 (12.9)	>0.99	13.6
Primary tumor type				
Thyroid	7 (9.5)	29 (19.7)	0.08	9.5
Salivary	13 (17.6)	24 (16.3)	0.96	17.6
Sarcoma	16 (21.6)	29 (19.7)	0.88	21.6
Lung	13 (17.6)	19 (12.9)	0.47	17.6
Other	25 (33.8)	46 (31.3)	0.82	33.7
Metastatic disease (vs. locally advanced, unresectable disease)	52 (96.3)	128 (87.1)	0.07	96.3
Central nervous system metastases (Yes)	16 (21.6)	18 (12.2)	0.10	21.6
*NTRK* gene fusion				
*NTRK1*	30 (40.5)	64 (43.5)	0.78	40.5
*NTRK2*	2 (2.7)	9 (6.1)	0.34	2.7
*NTRK3*	42 (56.8)	74 (50.3)	0.45	56.8
Prior lines of systemic therapy for metastatic disease				
0	20 (27.0)	39 (26.5)	>0.99	27.0
1	21 (28.4)	37 (25.2)	0.72	28.4
2	20 (27.0)	30 (20.4)	0.35	27.0
3+	13 (17.6)	41 (27.9)	0.13	17.6
Prior therapy (chemotherapy) ^c^	60 (81.1)	88 (59.9)	<0.01 *	81.1
				*p* < 0.01 *
Prior therapy (hormonal therapy) ^c^	9 (12.2)	6 (4.1)	<0.05 *	1.6
				*p* < 0.01 *
Prior therapy (immunotherapy) ^c^	9 (12.2)	16 (10.9)	0.95	15.0
				*p* = 0.61
Prior therapy (targeted therapy) ^c^	18 (24.3)	35 (23.8)	>0.99	19.6
				*p* = 0.46

^a^ Matching variables include the following: male, age above median in entrectinib population in Rolfo, 2020 (>57 years), White, Black, Asian, ECOG PS score 0, ECOG PS score 1, tumor (thyroid), tumor (sarcoma), tumor (salivary), tumor (lung), metastatic disease (vs. locally advanced, unresectable disease), central nervous system metastases (yes), *NTRK1*, *NTRK2*, prior lines of systemic therapy for metastatic disease (0), prior lines of systemic therapy for metastatic disease (1), prior lines of systemic therapy for metastatic disease (2). ^b^
*p*-values for continuous variables were calculated using the Wald test. *p*-values for categorical variables were calculated using the Wald test. ^c^ In the primary analysis, prior therapy type was not adjusted for. * denotes statistical significance (alpha < 0.05). ECOG: Eastern Cooperative Oncology Group; *NTRK*: neurotrophic receptor tyrosine kinase; PS: performance status.

**Table 3 cancers-14-01793-t003:** Comparison of efficacy outcomes before and after matching (primary analysis).

Time-to-Event Outcomes	Entrectinib	Larotrectinib Before Matching			Larotrectinib After Matching ^a^		
	Median, Months (95% CI)	Median, Months (95% CI)	HR vs. Entrectinib (95% CI)	*p*-Value	Median, Months (95% CI)	HR vs. Entrectinib (95% CI)	*p*-Value
OS	23.9	NR	0.43	<0.01	NR	0.43	<0.05 *
	(16.0, NE)	(40.7, NE)	(0.24, 0.76)		(38.7, NE)	(0.23, 0.83)	
PFS	11.2	33.0	0.56	<0.01	19.3	0.66	0.07
	(8.0, 15.7)	(16.6, NE)	(0.37, 0.86)		(11.5, 55.7)	(0.42, 1.03)	
DoR ^b^	12.9	41.5	0.33	<0.001	32.5	0.49	<0.05 *
	(9.3, NE)	(32.5, NE)	(0.17, 0.63)		(17.4, NE)	(0.25, 0.98)	
**Binary Outcomes**	**%**	**%**	**RD vs. entrectinib (95% CI)**	***p*-value**	**%**	**RD vs. entrectinib (95% CI)**	***p*-value**
	**(95% CI)**	**(95% CI)**			**(95% CI)**		
ORR	63.5	65.0	1.5	0.84	67.3	3.8	0.63
	(51.5, 74.4)	(56.1, 73.2)	(−12.5, 15.4)		(55.6, 77.2)	(−11.7, 19.3)	
CR	6.8	19.7	12.9	<0.01	20.3	13.5	<0.05 *
	(2.2, 15.1)	(13.2, 27.5)	(3.7, 22.1)		(12.8, 30.6)	(2.9, 24.1)	

CI: confidence interval; CR: complete response rate, DoR: duration of response, HR: hazard ratio, NR: not reached, ORR: overall response rate, OS: overall survival, PFS: progression-free survival, RD: risk difference, NE: Not estimable. * denotes statistical significance (alpha < 0.05). ^a^ Effective sample size = 72.71. ^b^ The sample size for DoR was 76 for larotrectinib and 32 for entrectinib.

**Table 4 cancers-14-01793-t004:** Comparison of safety outcomes before and after matching (primary analysis).

Variables	Entrectinib% (95% CI)	Larotrectinib Before Matching			Larotrectinib After Matching ^a^		
		% (95% CI)	RD vs. Entrectinib% (95% CI)	*p*-Value	% (95% CI)	RD vs. Entrectinib% (95% CI)	*p*-Value
Any serious TRAE	10.0	5.4	−4.6	0.27	6.3	−3.7	0.40
	(4.2, 20.1)	(2.5, 9.9)	(−12.6, 3.5)		(3.0, 12.8)	(−12.1, 4.8)	
TRAE leading to discontinuation	4.0	0.7	−3.3	0.18	0.7	−3.3	0.18
	(0.9, 12.4)	(0.0, 3.0)	(−8.2, 1.5)		(0.1, 4.6)	(−8.2, 1.5)	

CI: confidence interval; RD: risk difference; TRAE: treatment-related adverse event. ^a^ Effective sample size = 91.43.

**Table 5 cancers-14-01793-t005:** Treatment differences in the primary and sensitivity analyses.

Outcomes (Larotrectinib Relative to Entrectinib)	Primary Analysis	Sensitivity Analyses		
		Replacing Number of Lines of Prior Therapy with Type of Prior Therapy	Adding GI Tumors to the Matching Factors	Simulated Treatment Comparison
Overall survival, HR (95% CI)	0.43 (0.23, 0.83)	0.44 (0.23, 0.83)	0.44 (0.23, 0.84)	0.48 (0.27, 0.77)
Progression-free survival, HR (95% CI)	0.66 (0.42, 1.03)	0.58 (0.36, 0.93)	0.67 (0.42, 1.05)	0.76 (0.56, 1.15)
Overall response rate, RD (95% CI)	3.8 (−11.7, 19.3)	1.5 (−12.5, 15.4)	3.6 (−11.9, 19.1)	9.5 (−7.4, 26.4)
Complete response rate, RD (95% CI)	13.5 (2.9, 24.1)	12.9 (3.7, 22.1)	13.6 (3.0, 24.2)	18.2 (5.4, 30.9)
Duration of response, HR (95% CI)	0.49 (0.25, 0.98)	0.41 (0.20, 0.82)	0.50 (0.25, 0.98)	0.47 (0.24, 0.96)
Any serious TRAE, RD(95% CI)	−3.7 (−12.1, 4.8)	−6.0 (−13.9, 1.9)	−3.4 (−12.0, 5.2)	4.3 (−9.9, 18.5)
TRAE leading to discontinuation, RD (95% CI)	−3.3 (−8.2, 1.5)	−3.9 (−8.6, 0.7)	−3.3 (−8.2, 1.5)	−4.0 (−9.5, 1.5)

CI: confidence interval; HR: hazard ratio; RD: risk difference; TRAE: treatment-related adverse event; GI: gastrointestinal.

## Data Availability

Availability of the data underlying this publication will be based on Bayer’s commitment to the EFPIA–PhRMA Principles for responsible clinical trial data sharing. This pertains to scope, timepoint, and process of data access. As such, Bayer commits to sharing on request from qualified scientific and medical researchers patient-level clinical trial data, study-level clinical trial data, and protocols from clinical trials in patients for medicines and indications approved in the US and EU as necessary for doing legitimate research. This applies to data on new medicines and indications that have been approved by the EU and US regulatory agencies on or after 1 January 2014. Interested researchers can use www.clinicalstudydatarequest.com to request access to anonymized patient-level data and supporting documents from clinical studies to do further research that can help advance medical science or improve patient care. Information on the Bayer criteria for listing studies and other relevant information is provided in the study sponsors section of the portal. Data access will be granted to anonymized patient-level data, protocols, and clinical study reports after approval by an independent scientific review panel. Bayer is not involved in the decisions made by the independent review panel. Bayer will take all necessary measures to ensure that patient privacy is safeguarded.

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
