# Peer review of "Indirect Treatment Comparison of Larotrectinib versus Entrectinib in Treating Patients with TRK Gene Fusion Cancers"

_cancers, 2022, doi:10.3390/cancers14071793_

Round 1

Reviewer 1 Report

This is an interesting manuscript trying to assess and compare outcomes using an indirect Comparison of two available treatments (Larotrectinib versus Entrectinib) in patients with cancer types chracterized by TRK Gene Fusions.

Of course, such comparison (in the absence of a direct head-to-head comparison) remains suboptimal by common standards. However, given the rarity of these malignancies, such cross-study comparison may be of some interest for decision-making. The authors are very clear on this issue.

This reviewer suggests

  • to clarify the role of the company involved in design, analysis and writing of the study (results)
  • to more clearly describe comparisons in previous lines of treatment received and clarify differences in pre-treatment between the two groups
  • is remains confusing to this reviewer whether pts with CNS infiltration have been in-/excluded in one (both ?) arms ?
  • can information on length of treatment / cycles be provided ?
  • include a statement whether treatment in both arms was monotherapy
  • were there differences in sites invloved ? multicenter versus oligo-center study ?

Author Response

Response to reviewer 1

[Author responses in blue; additions to the manuscript are underlined]

Comments and Suggestions for Authors

This is an interesting manuscript trying to assess and compare outcomes using an indirect Comparison of two available treatments (Larotrectinib versus Entrectinib) in patients with cancer types chracterized by TRK Gene Fusions.

Of course, such comparison (in the absence of a direct head-to-head comparison) remains suboptimal by common standards. However, given the rarity of these malignancies, such cross-study comparison may be of some interest for decision-making. The authors are very clear on this issue.

This reviewer suggests

  • to clarify the role of the company involved in design, analysis and writing of the study (results)
    • Thank you, we added to the “Funding” section the following note:
      • “The study sponsor was involved in several aspects of the research, including the study design, the writing of the manuscript, and the decision to submit the manuscript for publication.”
  • to more clearly describe comparisons in previous lines of treatment received and clarify differences in pre-treatment between the two groups
    • Thank you for the suggestion, we added additional details about the differences between the two trials in section 3.1. Baseline characteristics:
      • “The proportion of patients without any prior lines of systemic therapy for metastatic disease is similar between the trials (26.5%, 27.0%, resp., p>0.99); fewer patients had 1 or 2 prior lines but more patients had 3+ lines in the larotrectinib trials compared to patients in the entrectinib trials (25.2% [larotrectinib] vs 28.4% [entrectinib], 20.4% vs 27.0%, and 27.9% vs 17.6%, resp.; all comparisons were non-significant).”
  • is remains confusing to this reviewer whether pts with CNS infiltration have been in-/excluded in one (both ?) arms ?
    • Thanks for the note. The analysis does not include patients with primary CNS tumors as the entrectinib trials did not include any patients with primary CNS, but patients may have had CNS metastases at baseline. We added two notes to clarify this point.
      • “This analysis did not include patients with primary central nervous system tumors as the efficacy population of the entrectinib trials did not include patients with primary central nervous system tumors.” (in section 2.2 Sample selection)
      • “While no patients with primary central nervous system tumors were present in the analysis, a total of 12.0% and 21.6% of patients in the respective trials had central nervous metastases at baseline (p=0.11).” (in section 3.1 Baseline characteristics)
  • can information on length of treatment / cycles be provided ? and include a statement whether treatment in both arms was monotherapy
    • Thank you, we added information on treatment for both studies in 2.1 Data sources.
      • “Larotrectinib was administered as monotherapy once or twice daily until either disease progression, withdrawal of the patient from the study, or the occurrence of an unacceptable level of adverse events.”
      • “Entrectinib was administered as monotherapy intermittently or once daily until either documented radiographic progression, unacceptable toxicity, or withdrawal of consent.”
  • were there differences in sites invloved ? multicenter versus oligo-center study ?
    • Thank you, all included trials were multicenter studies; we added notes to the descriptions where applicable in section 2.1 Data sources
      • SCOUT, a phase 1/2 multi-center study assessed the safety of larotrectinib (50-200mg) with advanced solid or primary CNS tumors [5].
      • NAVIGATE, a phase 2 multi-center basket study of larotrectinib (100mg) investigated its efficacy in the treatment of advanced sol-id tumors harboring a fusion of NTRK types 1-3 in children and adults [5].
    • Studies were either located in the US (larotrectinib’s LOXO-TRK-14001), Italy (entrectinib’s ALKA-372-001), or across multiple countries (all other studies). We added notes accordingly in section 2.1 Data sources
      • The combined larotrectinib trial population included patients recruited worldwide.
      • The combined entrectinib trial population included patients recruited worldwide.

Reviewer 2 Report

Very interesting comparisson of larotrectinib vs entrectinib in TRK fused solid tumors

I have only minor revisions

  1. I would like to understand more about the sample collection and wich way the matching was done. If the readers do not understand much of this, they will not be able to follow the results discribed in the article
  2. Why do you choose to analyse only these clinical trials, given that in the literature more trials are published on treatment of TRK fused tumors with larotrectinib or entrectinib, in order also to overcome the limitations of your study as you describe yourself?
  3. Did you compared baseline characteristics seaprately for the endpoints of your study?, for instance respone to larotectinib vs entrectinic according to tumor type, or prior lines of treatment or disease status (metastatic / locally advanced etc)?

Author Response

Response to reviewer 2

[Author responses in blue; additions to the manuscript are underlined]

Very interesting comparisson of larotrectinib vs entrectinib in TRK fused solid tumors

I have only minor revisions

  1. I would like to understand more about the sample collection and wich way the matching was done. If the readers do not understand much of this, they will not be able to follow the results discribed in the article
    • Thank you. Sample collection was described in section 2.1 Data sources and we added a note to section 2.4 Statistical analysis to clarify which data were used.
      • Patient-level data from larotrectinib and aggregate results from entrectinib (see Section 2.1) were used in the analysis.
    • We describe the matching process in Section 2.4, see description provided in the first bullet point below. We moved up the description of the analysis using matched data (shown in the second bullet point below) to make it easier for the reader to understand the matching process and subsequent analysis.
      • “Patients treated with larotrectinib were assigned weights so that the weighted average of selected baseline characteristics matched those of the entrectinib patient population. The weights were obtained based on a logistic regression model for the propensity of enrollment in the larotrectinib trials vs. the entrectinib trials. Because only summary data were available for the entrectinib trials, the logistic regression model was estimated using the method of moments [24].”
      • [Moved further up in the text] “After matching, baseline characteristics and outcomes of interest were compared between the weighted larotrectinib trial population and the entrectinib trial population using weighted t-tests for continuous variables and weighted chi-squared tests for categorical variables. Differences among continuous outcomes and risk differences (RDs) for categorical outcomes were calculated. Robust sandwich estimators were used to estimate standard errors, which provide valid estimates for 95% confidence intervals ([CIs]; i.e., achieved their nominal coverage probability). For time-to-event outcomes, survival curves were compared using weighted log-rank tests and hazard ratios (HRs) were estimated from weighted, otherwise unadjusted Cox proportional hazards models.”
  2. Why do you choose to analyse only these clinical trials, given that in the literature more trials are published on treatment of TRK fused tumors with larotrectinib or entrectinib, in order also to overcome the limitations of your study as you describe yourself?
    • Thank you for raising this point. The included studies are the respective pivotal trials for each drug and were therefore chosen for this comparison.
    • The methods we apply in this study require that included studies have sufficient overlap in their patient populations, which further limits the selection of trials to be included in the analysis.
  3. Did you compared baseline characteristics seaprately for the endpoints of your study?, for instance respone to larotectinib vs entrectinic according to tumor type, or prior lines of treatment or disease status (metastatic / locally advanced etc)?
    • Thank you. Unfortunately, we were not able to conduct comparisons on specific subgroups; the reason is that our analysis relied on published results from entrectinib and the required information for subgroups (baseline characteristics and results reported by subgroup) were not available. We note this limitation as follows in the manuscript in Section 5. Limitations and added more detail to address this:
    • “Second, the necessary information for an adjusted comparison (baseline characteristics and analysis results) were not reported by tumor type or other subgroups of interest for entrectinib, which prevented adjusted subgroup analyses. While the current study used the pooled population and matched on the tumor types, the adjustment could only be made for the most common tumor types due to the limited sample size.”